# Aromatase inhibition remodels the clonal architecture of estrogen-receptor-positive breast cancers

Christopher A. Miller[1,2], Yevgeniy Gindin[1,†], Charles Lu[1,†], Obi L. Griffith[1,3,4], Malachi Griffith[1,4,5], Dong Shen[1,†], Jeremy Hoog[3], Tiandao Li[1], David E. Larson[1,5], Mark Watson[6], Sherri R. Davies[3], Kelly Hunt[7], Vera J. Suman[8], Jacqueline Snider[3], Thomas Walsh[9], Graham A. Colditz[4,9], Katherine DeSchryver[3,9], Richard K. Wilson[1,2,3,4,5], Elaine R. Mardis[1,3,4,5] & Matthew J. Ellis[1,3,10]

Resistance to oestrogen-deprivation therapy is common in oestrogen-receptor-positive (ER +) breast cancer. To better understand the contributions of tumour heterogeneity and evolution to resistance, here we perform comprehensive genomic characterization of 22 primary tumours sampled before and after 4 months of neoadjuvant aromatase inhibitor (NAI) treatment. Comparing whole-genome sequencing of tumour/normal pairs from the two time points, with coincident tumour RNA sequencing, reveals widespread spatial and temporal heterogeneity, with marked remodelling of the clonal landscape in response to NAI. Two cases have genomic evidence of two independent tumours, most obviously an ER − 'collision tumour', which was only detected after NAI treatment of baseline ER + disease. Many mutations are newly detected or enriched post treatment, including two ligand-binding domain mutations in ESR1. The observed clonal complexity of the ER + breast cancer genome suggests that precision medicine approaches based on genomic analysis of a single specimen are likely insufficient to capture all clinically significant information.

[1] McDonnell Genome Institute, Washington University School of Medicine, St Louis, Missouri 63108, USA. [2] Department of Medicine, Division of Genomics and Bioinformatics, Washington University School of Medicine, St Louis, Missouri 63108, USA. [3] Department of Medicine, Division of Oncology, Washington University School of Medicine, St Louis, Missouri 63108, USA. [4] Siteman Cancer Center, Washington University School of Medicine, St Louis, Missouri 63108, USA. [5] Department of Genetics, Washington University School of Medicine, St Louis, Missouri 63108, USA. [6] Department of Pathology and Immunology, Washington University School of Medicine, St Louis, Missouri 63108, USA. [7] Department of Breast Surgery, MD Anderson Cancer Center, Houston, Texas 77030, USA. [8] Alliance Statistics and Data Center, Mayo Clinic, Rochester, Minnesota 55905, USA. [9] Department of Surgery, Division of Public Health, St Louis Breast Tissue Registry, Washington University School of Medicine, St Louis, Missouri 63108, USA. [10] Lester and Sue Smith Breast Center, Baylor College of Medicine, Houston, Texas 77030, USA. † Present addresses: Clontech Laboratories, 1290 Terra Bella Ave, Mountain View, California 94043, USA (Y.G.); AbbVie, 1401 Sheridan Rd, North Chicago, Illinois 60064, USA (C.L.); AstraZeneca, 1 MedImmune Way, Gaithersburg, Maryland 20878, USA (D.S.). Correspondence and requests for materials should be addressed to E.R.M. (email: emardis@wustl.edu) or to M.J.E. (email: mjellis@bcm.edu).

D espite the success of therapeutic approaches that suppress oestrogen production or inhibit ER function, resistance to endocrine therapy is common and accounts for the majority of breast cancer deaths[1]. The identification of endocrine resistance in primary ER+ breast cancer is a critical endeavour that has been widely explored[2,3] but the current transcriptional profiling approaches do not typically identify drivers of resistance, nor do they take into account resistance mechanisms that evolve from a minor subpopulation of cells.

Breast carcinomas are driven by accumulated somatic alterations. Those present in the initiating cell define the founder clone and persist in every cell of the tumour. As tumours grow, they accumulate additional mutations and these may give rise to subclonal populations with distinct characteristics[4,5]. These subpopulations compete and evolve[6–9] and often harbour mutations that eventually confer resistance to specific therapies[9–13]. When this occurs, the resistant tumour cells will represent a proportionally larger fraction of the tumour mass, as the susceptible population diminishes.

Another source of heterogeneity is synchronous breast cancer, most obviously when the disease is bilateral at diagnosis, or when two tumours are far apart in the same breast. These events are clinically significant as bilateral disease can signify the presence of a germline predisposition as well as a worse prognosis[14]. However when several independent breast cancers evolve in very close proximity, it can be impossible, without genomic techniques, to diagnose 'collision tumours' with different founder clones. The incidence of collision tumours of the breast and their clinical significance is therefore unknown, with most of the literature focused on tumours of different tissue origins growing together at a metastatic site[15].

While subclonal heterogeneity has been extensively studied in breast cancer[16,17], little is known about the interplay between clonal evolution and therapy response. The effects of short-term (14 day) aromatase inhibitor (AI) treatment have been explored on the transcriptomic level using microarrays[18,19], but it is unclear what population changes these expression differences reflect, that is, whether existing cells are being 'remodelled' at the transcriptional level, or whether the changes are due to continued expansion of cells harbouring resistance mutations. The effects of longer-term AI inhibition at the whole-genome level remain unknown.

In this study, we provide a first step towards answering these questions by comprehensively characterizing the effects of neoadjuvant aromatase inhibitor (NAI) therapy on the genomes and transcriptomes of a group of ER-positive breast tumours. From a set of previously whole-genome sequenced (WGS) tumours[20], we selected 22 Luminal A or Luminal B subtype tumours classified as either 'aromatase-inhibitor-sensitive' ($N = 12$, median surgical Ki67 = 1.1%, range 0–7.0%) or 'aromatase-inhibitor-resistant' ($N = 10$, median surgical Ki67 24.6%, range 10.4–47.1%). For each of these patients, we then performed WGS on matching tumours sampled by core biopsy at surgical resection, as well as on additional core biopsies taken from a subset of the baseline (pre-treatment, $N = 5$) and surgical resection tumours (post-treatment, $N = 6$). RNA sequencing was performed on 20 of the baseline tumours and on 18 of the surgical tumours. To obtain additional information on treatment emergent mutations, a subset of these tumours, along with 38 additional cases were analysed at greater sequencing depth using targeted capture with a gene panel. Together, these data allow us to characterize the genomic landscapes and clonal architectures of breast tumours and show that they are often dramatically altered during NAI therapy.

## Results

**Mutational landscape.** Comprehensive genomic characterization of these 22 patients' tumours (Table 1, Supplementary Data 1) revealed a total of 42,300 somatic single-nucleotide variants (SNVs) and indels in non-repetitive loci (tiers 1–3; ref. 21) of all tumour samples, with a median of 947 somatic SNVs and 23 indels per tumour (Supplementary Data 2 and 3). A total of 2,061 somatic

**Table 1 | Samples whole-genome sequenced in this study and a description of key clinical and biomarker parameters.**

| Case | Age | Intrinsic subtype | | Proportion cells Ki67 positive | | Ki67 response | ER allred | | Clonal instability index |
|------|-----|------|------|------|------|------|------|------|------|
| | | BL | SURG | BL | SURG | | BL | SURG | |
| BRC10 | 57 | LumB | LumA | 0.492 | 0.035 | Sensitive | 6 | 4 | 0.246 |
| BRC11 | 84 | LumB | LumB | 0.25 | 0.417 | Resistant | 7 | 6 | 0.097 |
| BRC14 | 86 | LumB | LumA | 0.442 | 0.012 | Sensitive | 7 | 4 | 0 |
| BRC15 | 83 | LumB | LumA | 0.238 | 0.01 | Sensitive | 7 | 7 | 0.579 |
| BRC17 | 63 | LumB | LumA | 0.32 | 0.019 | Sensitive | 7 | 8 | — |
| BRC18 | 85 | LumB | LumA | 0.125 | 0 | Sensitive | 8 | 7 | — |
| BRC20 | 61 | LumB | LumB | 0.456 | 0.349 | Resistant | 4 | 6 | 0.856 |
| BRC21 | 73 | LumA | Normal | 0.058 | 0.019 | Sensitive | 7 | 6 | 1 |
| BRC22 | 64 | LumA | LumA | 0.008 | 0 | Sensitive | 6 | 7 | 0.533 |
| BRC24 | 51 | LumB | LumB | 0.152 | 0.155 | Resistant | 7 | 7 | 0.403 |
| BRC26 | 71 | LumB | LumA | 0.101 | 0.07 | Sensitive | 7 | 5 | 0.499 |
| BRC30 | 60 | LumB | LumA | 0.256 | 0.183 | Resistant | 5 | 5 | 0.371 |
| BRC32 | 54 | LumA | LumA | 0.1 | 0 | Sensitive | 7 | 6 | 0.450 |
| BRC37 | 56 | LumB | LumB | 0.76 | 0.308 | Resistant | 6 | 4 | 0.359 |
| BRC38 | 78 | LumB | Her2 | 0.162 | 0.471 | Resistant | 8 | 2 | 1 |
| BRC39 | 79 | LumB | LumA | 0.356 | 0.124 | Resistant | 6 | 5 | 0.841 |
| BRC40 | 66 | LumA | LumA | 0.076 | 0.008 | Sensitive | 8 | 8 | 0.904 |
| BRC41 | 55 | LumB | LumA | 0.412 | 0.008 | Sensitive | 8 | 6 | 0.535 |
| BRC42 | 74 | LumA | LumA | 0.081 | 0.022 | Sensitive | — | 8 | 0.341 |
| BRC48 | 66 | LumA | LumA | ND | 0.346 | Resistant | 6 | 6 | — |
| BRC49 | 56 | LumA | LumA | 0.077 | 0.152 | Resistant | 8 | 8 | 0.715 |
| BRC50 | 78 | LumA | LumA | 0.195 | 0.104 | Resistant | 4 | 5 | 0.856 |

ND, not determined.
The clonal instability index is a metric of the amount of evolutionary change observed in the tumour, where higher numbers represent larger shifts and an index of 1 indicates completely unrelated tumours were present. A value of 0 indicates a monoclonal tumour with no post-treatment shift in clonality. Methodology to calculate the clonal instability index is defined in detail in the 'Methods' section. BL refers to baseline and SURG the pathology specimen from the surgical resection after 16 to 18 weeks of AI therapy.

variants were in the coding regions of the resulting proteins, with a median of 55 per patient. One tumour, BRC26, harboured a low mutation count (55), likely due to lower sequence coverage (Supplementary Data 4). Coding variant counts included 1,166 coding-region SNVs and indels not detected in the original study of the baseline tumours, obtained both from analysis of the surgical samples and via deeper sequencing coverage of baseline samples. Of these, 256 were absent or low frequency (<5% variant allele fraction (VAF) in the baseline samples, making them challenging or impossible to detect from standard-depth WGS without information from postsurgical or second core biopsy samples. The number of detectable low-frequency mutations was also impacted by tumour purity, which ranged from 12 to 100% (Supplementary Data 4). We identified 72 expressed coding mutations specific to the surgical samples, based on our limits of detection (Supplementary Note 1).

Copy number profiles were largely concordant between baseline and surgical samples, with a few notable exceptions (Supplementary Fig. 1). Eight samples harboured at least one event greater than 100 Mb in size that differed between the baseline and surgical samples, including one tumour (BRC48) that changed from diploid to triploid over the course of treatment. We also detected and validated a total of 1,695 structural variants (SVs) across 55 samples. The mean value was 30 SVs per sample (range 1 to 141) and had no significant correlation with either AI response or EFS (Supplementary Data 5). Overall, 71.8% of validated SVs present in the surgical samples were also detected in the baseline samples. Gene fusion-causing alterations were detected in 21 of 22 patients, with only

moderate concordance between multiple samples from the same individual (Supplementary Note 2, Supplementary Data 6). No fusions involving *ESR1* were observed. Transcriptome analysis revealed few differentially expressed genes and unsupervised hierarchical clustering did not identify clear expression signatures that distinguished AI-sensitive or -resistant tumours (Supplementary Figs 2–6, Supplementary Note 3, Supplementary Data 7 and 8).

**Subclonal inference by clustering of variant allele fractions.** Using somatic mutations from all the samples for each patient, we applied the SciClone algorithm[22] to cluster the VAFs of copy-number neutral SNVs, thus inferring the clonal architecture of each sample (Supplementary Fig. 7). Two samples, BRC17 and BRC48, had large-scale ploidy changes, which prevented accurate automated clustering of the clonal subpopulations, and a third case, BRC18, had a very high mutation rate and insufficient separation between clusters. The remaining 19 cases included 11 time points with multiple core biopsies. Several patterns of altered subclonal composition and evolution were observed in the context of AI treatment response, and are described below.

**Intertwined tumours of independent origin.** In one patient (BRC38), the surgical tumour shared no somatic SNVs or indels with the baseline tumour, despite having matching identity from germline SNP concordance (Fig. 1a). The ER+ baseline tumour contained a *CDH1* splice-site mutation in the founder clone, and

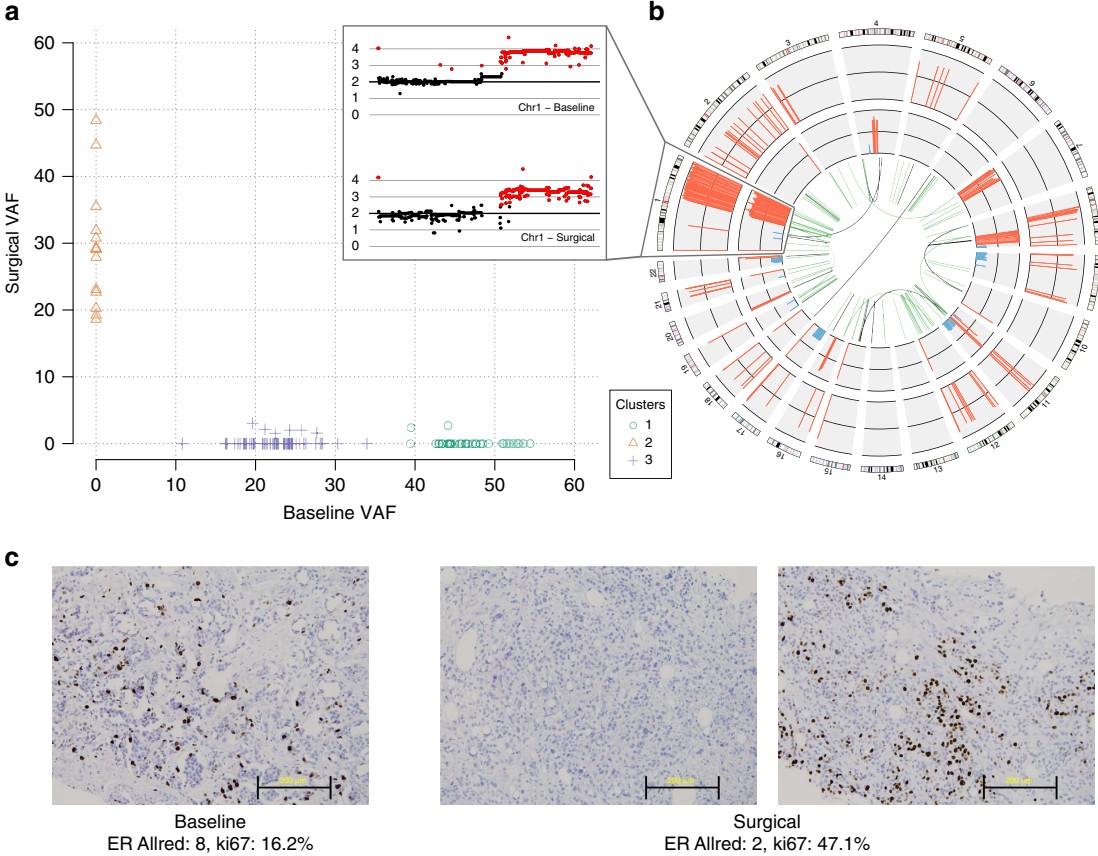

**Figure 1 | Collision tumours of independent origin and ER status in BRC38. (a)** Clonality plot comparing the VAFs of SNVs in the baseline and surgical samples. **(b)** Gene fusions and copy number alterations. Outer ring: CN alterations in the baseline sample (amplifications in red, deletions in blue). Inner ring: CN alterations in the surgical sample. Centre: gene fusion events that were specific to the baseline (green) or surgical sample (brown). Expansion: CN alterations on chromosome 1. **(c)** Immunohistochemistry results indicate the ER status of one baseline (left) and two surgical samples (middle, right). Scale bars, 200 μm.

subclonal missense mutations in *FOXA1* and *FOXQ1*. The surgical tumour specifically lacked these mutations or indeed other SNVs or indels in genes strongly implicated in cancer, but had focal amplifications containing *MYC*, *EGFR* and *CCND1* (Fig. 1b). Although the baseline tumour also had amplified *MYC* and *CCND1*, these were distinct events with breakpoints different than those observed in the surgical tumour. Both tumours contained numerous other copy number alterations, but none were shared except for a 1q amplification (with unresolvable breakpoints in the centromere). This event may have occurred in a shared cell of origin, or may be a case of homoplasy. Several gene fusions also were detected with no overlap between these two 'collision' tumours.

RNAseq data from this patient's tumour samples reveal a transition from baseline ER-positive (FPKM 131.65) to post-treatment ER-negative (FPKM of 0.99), and ER immunohistochemistry supported this observation, with ER Allred scores of 8 in the pre-treatment sample and 2 in the sequenced surgical sample (Fig. 1c–e, Supplementary Data 7). A second IHC result from a different portion of the post-treatment surgical sample showed residual ER-positivity, suggesting that the two tumours continued to co-exist, but that the surgical core we sequenced did not capture the ER-positive residual tumour. There was a 3-fold increase in Ki67 between baseline and surgery and this patient was classified as AI-resistant.

A second patient, BRC21, contained genomic evidence for a cryptic second tumour, inferred via a pattern of subclonal evolution that is impossible to explain with only a single tumour (Supplementary Fig. 7). Specifically, the variants in cluster 3 shifted from nearly absent at baseline to a VAF that was greater than any other cluster at surgery. This pattern suggests retention of the baseline tumour, coupled with the emergence of a second tumour containing a *PIK3CA* mutation (H1047R). These two tumours comprise 19 and 28 percent of the surgical sample, respectively, and this combined percentage is more parsimonious with the 70% estimate from pathology than would be a single-tumour solution with purity of less than 30%. Unlike BRC38, both appear to be ER+ tumours, with an Allred score at surgery of 6. Though both measures are limited somewhat by the low purity of the surgical sample, this patient was classified as AI-sensitive, and switched from an intrinsic subtype of LumA to Normal-like.

**Simple and clonally stable tumours**. Only a single sample, BRC14, harboured no detectable subclonal cell populations, either at baseline or following AI treatment (Fig. 2). This tumour was categorized as AI-sensitive. In keeping with AI responsiveness, the tumour intrinsic subtype switched from Luminal B to Luminal A.

**Complex and dynamic tumours**. Eighteen patients (81.8%) had tumours containing multiple subclonal cell populations, some of which were substantially altered during the course of treatment. In BRC15, the two baseline samples had similar clonal composition, but the two surgical samples contained significant spatial heterogeneity as well as extensive remodelling of the clonal architecture (Fig. 3, Supplementary Note 4). This tumour had over 2.5-fold higher *ESR1* mRNA expression levels at surgery than at baseline and was sensitive to aromatase-inhibitor treatment as determined by the drop in Ki67 level from 24 to 1%.

Four-dimensional clustering revealed a founder clone and four subclones. The founder clone contained expressed mutations in *PIK3CA* (H1047R) and *ARID2* (frameshift insertion). Cluster two was the dominant subclone in the pre-treatment sample and post-treatment sample 2, but comprised only about 11% of post-treatment sample 1. Cluster three, which contained expressed mutations in *BRAF* (K601E) and *MSH6* (nonsense), made up 92% of the baseline tumour, but was absent in the post-treatment samples, suggesting that it harboured enhanced susceptibility to oestrogen deprivation. Cluster four was undetectable in the baseline samples (limit of detection ~1% VAF), but constituted 89% of post-treatment core 2, where clonal expansion may have been driven by a second mutation in *PIK3CA* (G118D). Similarly, cluster 5 was absent pre-treatment but present only in post-treatment core 1. The copy number profiles between all samples were largely concordant, with notable differences including amplification of chromosome 8 in surgical sample 1 and deletion of one copy of chromosome 9 in surgical sample 2. Three samples from the periphery of this tumour (all classified as Ductal Carcinoma *in situ*) were obtained from the surgical formalin-fixed block; all carried the founder clone *BIRC6* and *PIK3CA* mutations, and one carried the *BRAF* mutation found in cluster 3. Additional descriptions of these sample genotypes can be found in the Supplementary Note 5 and Supplementary Data 8.

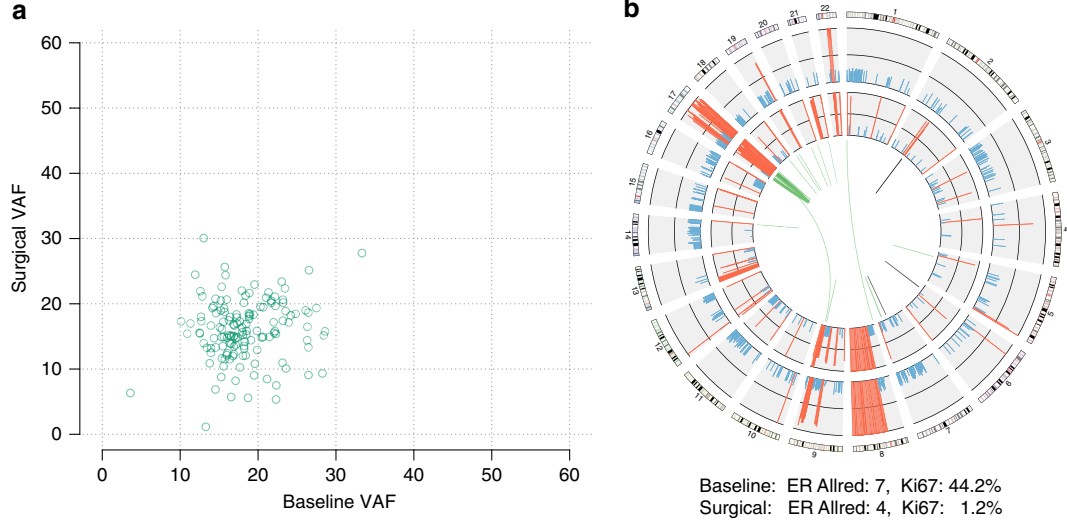

Baseline: ER Allred: 7, Ki67: 44.2%
Surgical: ER Allred: 4, Ki67: 1.2%

**Figure 2 | Simple and stable clonal structure in BRC14.** (**a**) Clonality plot comparing the variant allele fraction of SNVs in the baseline and surgical samples. (**b**) Gene fusions and copy number alterations. Outer ring: CN alterations in the baseline sample (amplifications in red, deletions in blue). Inner ring: CN alterations in the surgical sample. Centre: gene fusion events that were specific to the baseline (green) or surgical sample (brown).

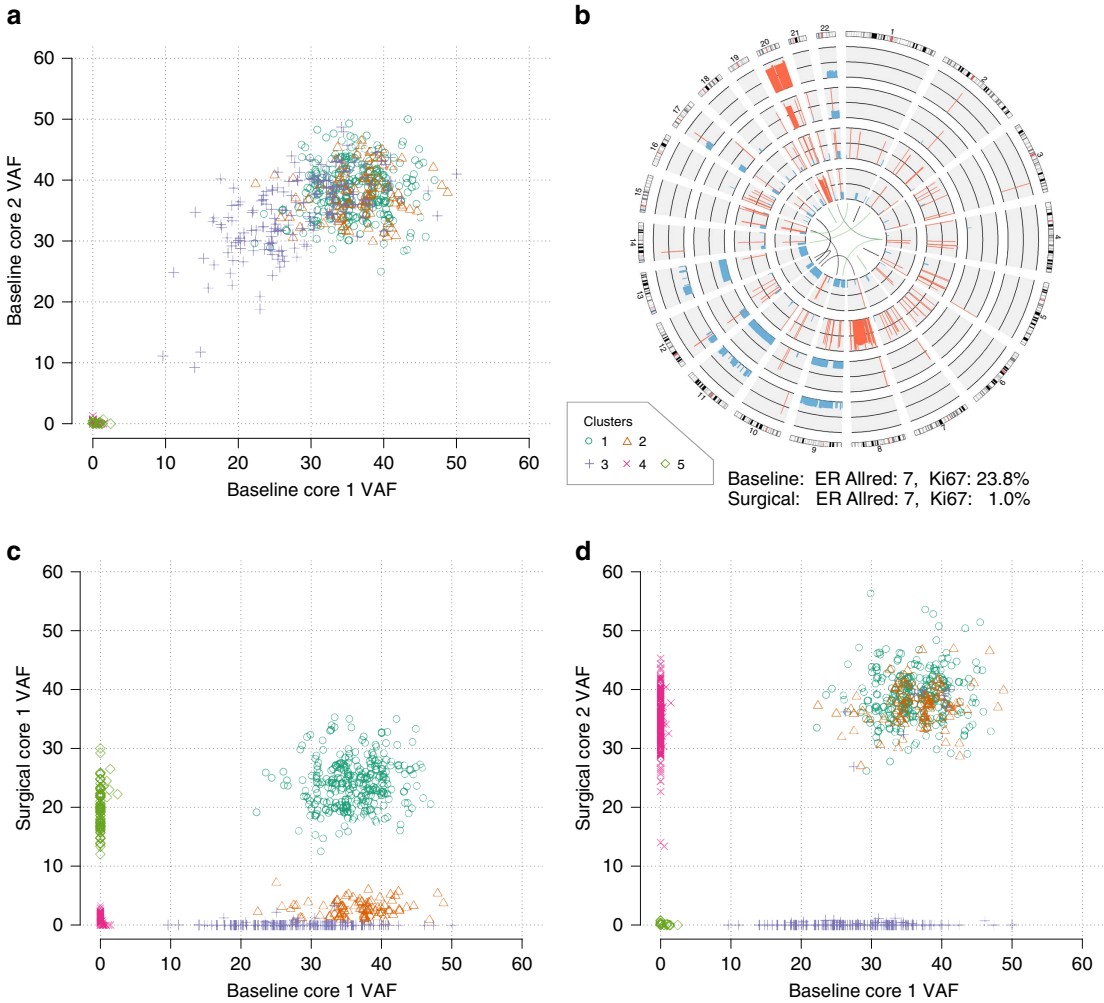

**Figure 3 | Subclonal complexity and response to AI inhibition in BRC15.** Clonality plots derived from four-dimensional clustering of SNV VAFs in distinct core samples. (**a**) Two samples separated spatially in the baseline tumour. (**c,d**) The first baseline core sample compared with two cores taken from the surgical sample. (**b**) Gene fusions and copy number alterations in (from outer ring to inner ring) Baseline core 1, Baseline core 2, Surgical core 1, Surgical core 2 (amplifications in red, deletions in blue). Centre: gene fusion events that are baseline sample-specific (green), surgical sample-specific (brown) or shared (black).

BRC41 provides another striking example of tumour evolution under oestrogen deprivation, with almost complete loss of a subclone that comprised 86% of the baseline tumour cells (Supplementary Fig. 7). This subclone loss was coincident with the emergence of a rare subclone that was present in about 2% of baseline tumour cells, but expanded to 84% of the surgical sample. The responsive baseline subclone included an expressed *KDM8* (R227H) histone demethylase mutation. The emergent resistant subclone contained an expressed mutation in *BCAS3* (K533N), a gene with some evidence linking it to tamoxifen resistance[23]. Unlike BRC15, the paired core samples taken at baseline and at surgery were highly concordant, suggesting minimal spatial heterogeneity in this tumour.

In addition to these two cases, 13 other tumours were computationally classified as clonally complex and dynamic (Supplementary Fig. 7). To do so, we used the mean absolute difference in clonal fractions before and after treatment to derive a clonal instability index, where 0 indicates no change in clonal architecture and 1 indicates completely distinct tumours (see 'Methods' section, Supplementary Data 10). Among the 15 patients in this complex and dynamic group, we observed clonal instability values ranging from 0.25 to 0.90 (median 0.53), reflecting the large degree of temporal change in clonality during

NAI treatment. Three additional cases were manually assigned to this category, for a total of 18 complex and dynamic tumours. Of these, 10 were characterized as responsive and 8 were resistant to AI. There was no significant correlation between clonal instability and AI response, as measured by either reduction in Ki67 or Ki67 at surgery. We observed intrinsic subtype switching in nine of these samples, with all changing from Luminal B to Luminal A.

**Complex and stable tumours**. Only one sample (BRC11) had detectable subclonal populations, but showed little evidence of treatment-induced remodelling, with a clonal instability score of 0.097 (Fig. 4). The likely founder clone carried missense mutations in *ARID1A* (p.A1239T), *PIK3CA* (p.H1047R) and a frame-shift deletion in *TP53*. The tumour was classified as treatment-resistant with Ki67 rising from 0.25 to 0.417 at surgery.

**Targeted sequencing and treatment-associated heterogeneity**. To examine AI treatment-associated changes at the level of a sequencing panel typical of clinical NGS-based diagnostic assays, we developed a hybrid capture panel to sequence 83 breast cancer-related genes in 19 of the WGS cases, plus an additional 38

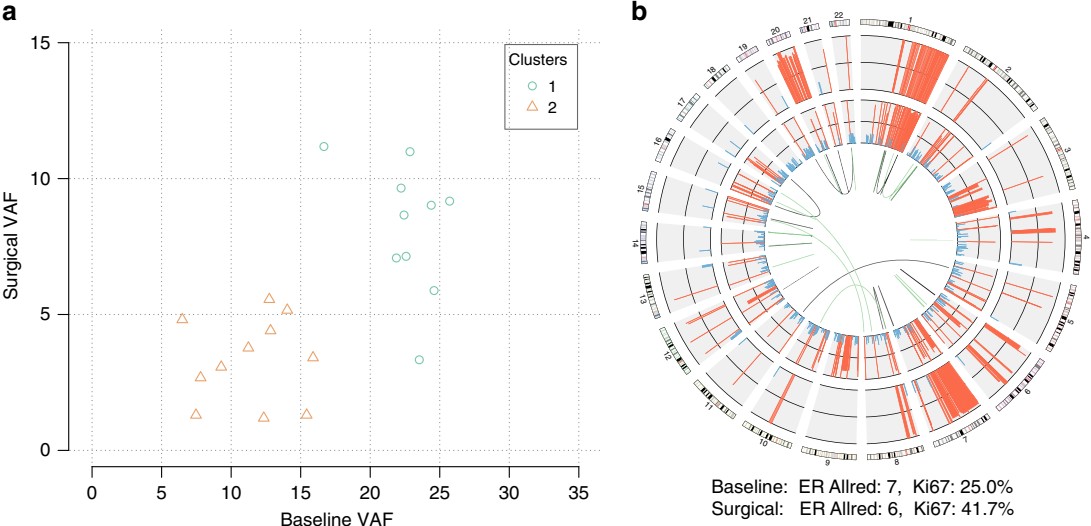

**Figure 4 | Complex and stable clonal structure in BRC11.** (**a**) Clonality plot comparing the variant allele fraction of SNVs in the baseline and surgical samples. (**b**) Gene fusions and copy number alterations. Outer ring: CN alterations in the baseline sample (amplifications in red, deletions in blue). Inner ring: CN alterations in the surgical sample. Centre: gene fusion events that were specific to the baseline (green) or surgical sample (brown).

pre/post pairs (Table 2, Supplementary Note 5, Supplementary Data 11 and 12). Of these 57 sample sets, 38 (from both groups) had paired baseline and surgical samples as described above, the other 19 had samples from baseline and end of treatment (most of which were triaged to chemotherapy). The mean depth of coverage was ×103.4 in this assay, compared with a mean of ×44.1 in the WGS and ×79.4 from the combined WGS and validation sequencing on the original 22 patients

We identified a total of 255 mutations from this panel test (median 3 per patient), three of which occur in genes that are high-priority drug targets. The *ESR1* D538G mutation is a well-recognized ligand-binding domain (LBD) mutation, associated with ligand-independent transcriptional activity[24,25] and was present only in the post-treatment sample (11% VAF). The low post-treatment Ki67 suggests, however, that at 11% the *ESR1* D538G allele was not yet sufficiently dominant to produce elevated proliferation despite AI treatment. The second treatment-emergent LBD mutation was K481N, present at 14% VAF at baseline and ~40% VAF post treatment. The increasing VAF of this K481N mutation was associated with Ki67-defined resistance, since the Ki67 was 11% in the post-treatment sample, sufficiently high to elevate the risk of relapse[26]. A third LBD mutation, E380Q, was present at a low frequency at baseline without enrichment suggesting that this mutation does not strongly confer AI resistance, as suggested by an earlier study of ER+ patient-derived xenografts[17].

To determine whether somatic mutations in particular genes were driving clonal response, we calculated the difference between pre-treatment and post-treatment VAFs for each gene with at least five non-silent mutations (Supplementary Fig. 3). Though *NCOR1* mutations, in particular, trended towards higher VAFs in the post-treatment samples, a larger cohort would be needed to establish significance in the context of multiple testing correction. There also was no significant difference in VAF shifts between mutations in different *PIK3CA* hotspots (near amino acids 1,047 or 543) or between mutations in different TP53 domains (DNA-binding versus other).

## Discussion

In this study, we performed deep genomic characterization of primary ER+ breast tumours at baseline and after 16 to 18 weeks

of AI therapy. By combining genomic sequencing data at high depth on mutated sites, copy number and structural variant discovery, comparisons of RNA expression to derive intrinsic subtypes, and clinical pathology markers such as Ki67 and ER Allred scores, we have generated comprehensive information about the range of changes that occur when ER+ breast cancers are subjected to oestrogen deprivation. Overall, four genomic patterns were observed: (1) two intertwined but genomically separate 'collision tumour' patterns; (2) 'clonally simple and treatment stable' patterns; (3) 'clonally complex and treatment dynamic' patterns; and (4) 'clonally complex and treatment stable' patterns.

In two cases, we observed collision tumours comprised separate malignancies, and in one case an ER+ tumour with indolent clinical features was replaced by an ER− tumour with aggressive clinical features. Presumably the ER− tumour genotype was missed in the pre-treatment sample, and became more readily detectable over time due to the regression of the ER+ tumour. It is well recognized that a subset of ER+ breast cancers have 'lost' ER expression upon relapse[27], and earlier studies report that ER loss in the context of neoadjuvant endocrine therapy is associated with a much higher risk of relapse[26]. Together, these suggest that comparative ER testing should be repeated on residual breast tumours after NAI therapy to identify discernable changes in ER positivity.

Clonally stable tumours were rare in this cohort, with only one simple and one complex tumour. The simple tumour (BRC14) switched from Luminal B to Luminal A with a dramatic decline in Ki67 (44.2 to 1.2%). The complex tumour (BRC11) likely contained innate resistance in the founder clone (with no evidence of sensitivity to the antiproliferative effects of AI), which enabled it to maintain both its clonal structure and its intrinsic subtype despite treatment.

Most tumours analysed (18 of 22) were clonally heterogeneous and contained subclonal populations whose relative proportions changed, often dramatically, during AI treatment. This high clonal instability, over 4 months of AI treatment, was likely due to a selective growth advantage for a subclone in the presence of oestrogen deprivation. While it is hard to disprove the null hypothesis, that is, we are simply observing spatial heterogeneity in a very complex tumour, cases like BRC41 displayed very high levels of temporal change with little difference between spatially

**Table 2 | Thirty-eight additional cases sequenced with an 83-gene panel and a description of key clinical and biomarker parameters.**

| Case | Age | PT time point | Intrinsic subtype | | Proportion cells Ki67 positive | | Ki67 response | ER Allred | |
|---|---|---|---|---|---|---|---|---|---|
| | | | BL | PT | BL | PT | | BL | SURG |
| 586120 | 68 | EOT | LumB | LumA | 0.299 | 0.019 | Sensitive | 7 | 8 |
| 439295 | 58 | Surgery | LumB | LumB | 0.53 | 0.453 | Resistant | 7 | 6 |
| 228281 | 85 | EOT | LumB | LumA | 0.387 | 0.045 | Sensitive | 8 | 7 |
| 412952 | 81 | Surgery | LumB | Normal | 0.192 | 0.013 | Sensitive | 6 | 6 |
| 427207 | 78 | Surgery | LumB | LumB | 0.426 | 0.196 | Resistant | 7 | 6 |
| 687744 | 55 | Surgery | LumB | LumB | 0.074 | 0.021 | Sensitive | 5 | 4 |
| 251582 | 56 | Surgery | LumA | NA | 0.049 | 0.003 | Sensitive | 7 | 7 |
| 401301 | 63 | Surgery | Her2 | Her2 | 0.379 | 0.343 | Resistant | 4 | 0 |
| 949339 | 53 | Surgery | LumA | LumA | 0.003 | 0.001 | Sensitive | 5 | NA |
| 148037 | 62 | Surgery | LumB | NA | 0.385 | 0.336 | Resistant | 8 | 6 |
| 394713 | 64 | EOT | LumB | LumA | 0.489 | NA | Resistant | 7 | NA |
| 204983 | 62 | EOT | LumB | LumA | 0.217 | NA | Resistant | 8 | NA |
| 144029 | 74 | Surgery | LumB | Her2 | 0.487 | 0.224 | Resistant | 6 | 2 |
| 396695 | 54 | Surgery | LumA | NA | 0.265 | 0.002 | Sensitive | 7 | 8 |
| 229684 | 58 | Surgery | Basal | Basal | 0.388 | 0.268 | Resistant | 3 | 0 |
| 808150 | 55 | Surgery | LumB | Normal | 0.154 | 0.077 | Sensitive | 8 | 7 |
| 755730 | 62 | Surgery | LumB | LumA | 0.177 | 0.009 | Sensitive | 7 | 4 |
| 895779 | 90 | EOT | LumB | LumB | 0.327 | 0.015 | Sensitive | 8 | 7 |
| 228281 | 66 | Surgery | LumB | LumA | 0.136 | 0.03 | Sensitive | 7 | 7 |
| 314722 | 73 | Surgery | LumA | LumA | 0.112 | 0.029 | Sensitive | 7 | 4 |
| 641677 | 62 | Surgery | HER2 | Her2 | 0.695 | 0.515 | Resistant | 6 | 3 |
| 520102 | 54 | EOT | LumA | LumA | 0.261 | 0.006 | Sensitive | 8 | 7 |
| 948809 | 58 | Surgery | LumB | LumA | NA | 0.043 | Sensitive | 6 | 7 |
| 982661 | 53 | EOT | LumA | LumA | 0.127 | 0.01 | Sensitive | 7 | 7 |
| 832844 | 59 | EOT | LumB | LumA | 0.28 | 0.21 | Resistant | 8 | 8 |
| 287368 | 66 | EOT | Her2 | Her2 | 0.9 | 0.238 | Resistant | 7 | 8 |
| 411144 | 76 | EOT | LumA | LumB | 0.091 | 0.162 | Resistant | 7 | 7 |
| 702554 | 55 | EOT | LumB | LumB | 0.6 | 0.8 | Resistant | 7 | 7 |
| 251239 | 73 | EOT | LumB | LumB | 0.3 | 0.063 | Sensitive | 7 | 7 |
| 971640 | 53 | Surgery | LumB | Normal | 0.4 | 0.003 | Sensitive | 7 | 8 |
| 963465 | 55 | Surgery | LumB | NA | 0.121 | 0.015 | Sensitive | 8 | 8 |
| 553787 | 63 | EOT | LumB | NA | 0.251 | 0.186 | Resistant | 8 | 8 |
| 451180 | 47 | EOT | LumB | LumB | 0.3 | 0.125 | Resistant | 8 | 8 |
| 306707 | 67 | EOT | LumB | LumB | 0.266 | 0.117 | Resistant | 7 | 7 |
| 147888 | 64 | EOT | LumB | LumB | 0.398 | 0.274 | Resistant | 8 | 7 |
| 768794 | 57 | EOT | LumB | NA | 0.192 | 0.17 | Resistant | 8 | 8 |
| 629051 | 65 | EOT | LumA | LumA | 0.086 | 0.01 | Sensitive | 8 | 8 |
| 625428 | 51 | EOT | LumB | LumA | 0.5 | 0.252 | Resistant | 8 | 8 |

NA, not available.
BL refers to baseline and PT the pathology specimen from either the surgical resection after 16 to 18 weeks of therapy or the time at which the patient was removed from the trial (EOT). All the cases in Table 1 were also assayed on this panel, with the exceptions of BRC11, BRC20 and BRC48.

separate cores taken concurrently. This evidence supports the argument for tumour evolution under selective pressure as the likely explanation. On the other hand, 63% of the multiple concurrent biopsies that we sequenced exhibited spatial heterogeneity. Thus, the shifts in clonal architecture that we observed are likely due to a combination of therapy-related subclonal selection and sampling of different subclonal populations within the tumour at different time points.

Few resistance mechanisms were revealed by this study, as only two *ESR1* mutations were treatment emergent at the 16 to 18 week time point and no *ESR1* translocations were detected by WGS or RNA seq[17]. This suggests that *ESR1* mutation-driven resistant subclones emerge over long periods of time, although the presence of several easily detectable *ESR1* mutations at ~100X coverage suggests that deeper sequencing might be important to detecting these mutations at an early-on treatment time point. The ALTERNATE neoadjuvant endocrine therapy trial is comparing an AI with fulvestrant, an ER-degrading agent (NCT01953588). The sample size for ALTERNATE (over 1,000 patients) and the longer duration of treatment before surgery

(24 weeks) will eventually allow hypotheses based on detecting low frequency *ESR1* mutant alleles in the post-treatment sample to be explored more thoroughly, and will determine whether the choice of endocrine agent influences the rate at which *ESR1* mutations evolve.

Interestingly, two tumours were diagnosed with activating *ERBB2* mutations[28], however, both were in AI-sensitive tumours, with very low on-treatment KI67 values. This suggests, at least anecdotally, that *ERBB2* activating mutations do not cause intrinsic endocrine therapy resistance, with high on-AI-treatment Ki67 values that were noted when studying HER2 amplified ER + tumours in the neoadjuvant endocrine setting[29]. One other treatment emergent mutation in *ERBB2* was noted, S157F, but the functional significance of the mutation is uncertain. A final point about mutation-driven therapeutic hypotheses concerns the pre- and post-treatment *PIK3CA* data. It has been previously reported that there is no interaction between *PIK3CA* mutation status and response to neoadjuvant endocrine therapy[20,30]. The lack of positive or negative selection for *PIK3CA* mutations observed in the present study is compatible with this conclusion. Of the three tumours with *PIK3CA* mutations detected at surgery,

but not at baseline, two were collision tumours, where the second tumour containing mutant *PIK3CA* was missed by the initial biopsies. Thus, spatial heterogeneity explains this pattern, and also is the likely cause of the apparent gain of a G118D mutation in one of the BRC15 post-treatment samples. This is supported by the lack of trend toward positive or negative selection in the 16 kinase domain mutant cases studied herein (Table 3).

In conclusion, caution needs to be exercised when interpreting clinical genomic results from a single core biopsy at a single time point. Higher depth of massively parallel sequencing, coupled with analysis tools to detect low-VAF variants, analysis of multiple tumour samples and tracking of genomic evolution in post-therapy samples all provide a more complete picture. These are feasible techniques, as we illustrate, and the data we report can be considered a proof of principle, supporting the idea that investment is needed to collect and study very large cohorts in such a manner. These studies will ultimately determine the clinical utility of genomic analysis in ER+ breast cancer that is more comprehensive than that afforded by analysis of a single pre-treatment sample.

## Methods

**Sample acquisition and clinical characterization.** The samples were from two neoadjuvant endocrine therapy trials of postmenopausal women with clinical stage II to III ER-positive (Allred score 6–8) breast cancer (ACOSOG-Z1031, Alliance, NCT00265759 and NCT00084396) that have been previously described[31,32]. All the samples were studied that had appropriate consents, at least 70% tumour content (by nuclei), and available DNA from both the baseline and surgical time points.

**Genomic library preparation and sequencing.** Library preparation was performed on DNA extracted from flash-frozen fresh tissue biopsies and matched peripheral blood, then sequenced. Both were done as described previously[20].

**Copy number and structural variant detection.** Copy number aberrations were detected using CopyCat v1.6.9 (https://github.com/chrisamiller/copycat) (Supplementary Software 2), with purity estimates derived from SNV allele fractions. (Table 1) Structural variants were detected as previously described[20,33]. For measurement of concordance, SVs were considered to be the same event if the ends of each call were localized within 500 bp. Gene fusions were identified using INTEGRATE v0.1e, with default parameters[34].

**SNV and indel detection.** We detected SNVs using the union of three callers: (1) samtools version r963 [2] (params: -A -B) filtered by SNP-filter version v1 and intersected with Somatic Sniper version 1.0.2 [3] (params: -F vcf -q 1 -Q 15) filtered by false-positive version v1 (params: --bam-readcount-version 0.4 --bam-readcount-min-base-quality 15) then somatic-score-mapping-quality version v1 (params:--min-mapping-quality 40 --min-somatic-score 40); (2) VarScan 2.2.6 [4] filtered by varscan-high-confidence version v1 then false-positive version v1 (params: --bam-readcount-version 0.4 --bam-readcount-min-base-quality 15); (3) Strelka version 0.4.6.2 [5] (params: isSkipDepthFilters = 0).

Short insertions and deletions were detected using the union of four callers: (1) gatk-somatic-indel version 5336 [6] filtered by false-indel version v1 (params: --bam-readcount-version 0.4 --bam-readcount-min-base-quality 15); (2) pindel version 0.5 [7] filtered by pindel-somatic-calls version v1 then pindel-vaf-filter version v1 (params: --variant-freq-cutoff = 0.08) then pindel-read-support version v1; (3) VarScan 2.2.6 [6] filtered by varscan-high-confidence-indel version v1 then false-indel version v1 (params: --bam-readcount-version 0.4 --bam-readcount-min-base-quality 15); (4) Strelka version 0.4.6.2 [5] (params: isSkipDepthFilters = 0).

For SNVs and Indels in tier 2 and 3, we retained all variant calls that were also called in the deep validation data. Additional steps were taken to enhance the number of coding variants recovered, including pooling of discovery (WGS) and validation (capture) data to provide enhanced coverage. A binomial log-likelihood

## Table 3 | Druggable mutations observed in samples assayed with targeted sequencing.

| Sample | WGS | Gene | Amino acid change | Baseline VAF | Surgical VAF | Ki67 BL | Ki67 PT | Ki67 response |
|---|---|---|---|---|---|---|---|---|
| 451180 | — | ERBB2 | p.R599C | 4.59 | 0 | 0.3 | 0.125 | Resistant |
| 917386 | BRC14 | ERBB2 | p.V777L | 26.67 | 31.55 | 0.442 | 0.012 | Sensitive |
| 148037 | — | ERBB2 | p.780in_frame_insGSP | 2.67 | 12.35 | 0.003 | 0.001 | Sensitive |
| 963465 | — | ERBB2 | p.S157F | 0 | 7.21 | 0.121 | 0.015 | Sensitive |
| 451180 | — | ESR1 | p.E380Q | 1.19 | 3.9 | 0.3 | 0.125 | Resistant |
| 306707 | — | ESR1 | p.K481N | 14.17 | 39.71 | 0.266 | 0.117 | Resistant |
| 375938 | BRC22 | ESR1 | p.D538G | 0 | 11.93 | 0.008 | 0 | Sensitive |
| 255394 | BRC39 | PIK3CA | p.P104R | 52.73 | 23.81 | 0.356 | 0.124 | Resistant |
| 702554 | — | PIK3CA | p.N345K | 29.41 | 34.09 | 0.6 | 0.8 | Resistant |
| 229684 | — | PIK3CA | p.G364R | 34.33 | 29.49 | 0.265 | 0.002 | Sensitive |
| 629051 | — | PIK3CA | p.E418K | 35.06 | 41.54 | 0.086 | 0.01 | Sensitive |
| 228281 | — | PIK3CA | p.C420R | 22.58 | 29.71 | 0.327 | 0.015 | Sensitive |
| 169316 | BRC38 | PIK3CA | p.E542K | 0 | 28.95 | 0.162 | 0.471 | Resistant |
| 434673 | BRC42 | PIK3CA | p.E542K | 10.17 | 4.55 | 0.081 | 0.022 | Sensitive |
| 144029 | — | PIK3CA | p.E545K | 16.67 | 28.57 | 0.217 | NA | Resistant |
| 306707 | — | PIK3CA | p.E545K | 10.68 | 38.05 | 0.266 | 0.117 | Resistant |
| 629051 | — | PIK3CA | p.E545Q | 39.58 | 29.89 | 0.086 | 0.01 | Sensitive |
| 228281 | — | PIK3CA | p.E545K | 84.13 | 26.28 | 0.387 | 0.045 | Sensitive |
| 412952 | BRC39 | PIK3CA | p.N1044K | 41.11 | 19.75 | 0.356 | 0.124 | Resistant |
| 255394 | BRC21 | PIK3CA | p.H1047R | 0 | 19.28 | 0.058 | 0.019 | Sensitive |
| 441655 | BRC40 | PIK3CA | p.H1047R | 27.18 | 27.14 | 0.076 | 0.008 | Sensitive |
| 956936 | BRC37 | PIK3CA | p.H1047R | 55.52 | 66.25 | 0.76 | 0.308 | Resistant |
| 303279 | BRC10 | PIK3CA | p.H1047R | 50 | 36.27 | 0.492 | 0.035 | Sensitive |
| 632762 | — | PIK3CA | p.H1047R | 48.84 | 19.21 | 0.388 | 0.268 | Resistant |
| 808150 | BRC50 | PIK3CA | p.H1047L | 48.26 | 45.18 | 0.195 | 0.104 | Resistant |
| 629834 | — | PIK3CA | p.H1047R | 60 | 33.73 | 0.177 | 0.009 | Sensitive |
| 895779 | — | PIK3CA | p.H1047R | 12.04 | 21.77 | 0.112 | 0.029 | Sensitive |
| 641677 | BRC15 | PIK3CA | p.H1047R | 37.78 | 27.17 | 0.238 | 0.01 | Sensitive |
| 384803 | BRC24 | PIK3CA | p.H1047R | 35.63 | 24.21 | 0.152 | 0.155 | Resistant |
| 526430 | BRC18 | PIK3CA | p.H1047R | 23.64 | 23.58 | 0.125 | 0 | Sensitive |
| 767881 | — | PIK3CA | p.H1047R | 54.29 | 32.07 | 0.091 | 0.162 | Resistant |
| 411144 | — | PIK3CA | p.H1047R | 22.35 | 2.99 | 0.3 | 0.063 | Sensitive |
| 251239 | — | PIK3CA | p.H1047R | 45.65 | 45.45 | 0.53 | 0.453 | Resistant |
| 439295 | — | PIK3CA | p.G1049R | 23.53 | 22.16 | 0.154 | 0.077 | Sensitive |

filter (https://github.com/genome/genome/blob/master/lib/perl/Genome/Model/Tools/Validation/IdentifyOutliers.pm) was applied to variant calls with at least $10\times$ sequence coverage in both tumour and normal samples, and those identified as somatic with LLR > 3 were retained. Review of mutational hotspots was used to recover additional variants in breast cancer-related genes.

**Variant validation.** Mutations in the 22 WGS samples from the original paper were validated using custom capture arrays targeted to the mutations in each tumour, as previously described[20]. After sequencing the post-AI samples, a second round of custom capture validation was performed on all 77 samples from these 22 patients, by combining the somatic mutation sites predicted in both the baseline and surgical samples. All coding SNVs, indels and SVs were targeted for validation, along with a random subset of non-coding variants, providing additional sequence coverage for validation and clonality analysis. No Sanger sequencing validation was performed. Controlled tests of our somatic mutation calling pipeline (with similar configuration to that used here) have established a positive predictive value of over 90% for mutations down to VAFs of ~5%, so we expect the baseline false-positive rate to be low in these data. Results from the validation sequencing pipeline were then subjected to extensive manual review to validate the presence of variants and remove additional false-positive somatic calls. This involved manually inspecting sequencing reads using the Integrated Genomic Viewer (IGV), and flagging a number of different types of sequencing artifacts, including: insertions and deletions at the ends of homopolymer runs, calls that were present in the matched normal sample (or other normal samples), and regions with substantial numbers of low-or-zero quality mapping scores, indicating potential reference build problems or mapping artifacts. A similar process was used to inspect copy number calls, removing those resulting from alignment errors in highly repetitive regions of the genome or that span assembly gaps or centromeres.

**Baseline and surgical-specific variants.** Variant allele fractions of all tier 1 variants were corrected for purity by reducing the number of reference-supporting reads in proportion to the purity of the sample. This effectively scales the VAFs in such a way that founding clone variants are near 50% VAF. At each variant position, Fisher's exact test was used to identify significant VAF changes, by comparing each baseline sample with each surgical sample. Those with $P < 0.05$ were retained. Baseline-specific variants were required to be below 10% VAF in the surgical sample and have at least one variant-supporting read in the baseline RNA sample. Surgical-specific variants were required to be below 10% VAF in the baseline sample and have at least one variant-supporting read in the surgical RNA sample.

**Targeted sequencing of extension cohort.** A panel of 83 breast cancer-related genes was derived from meta-analysis and literature review (Supplementary Data 11). These genes were targeted comprehensively with 3,029 complementary probes for hybridization-based enrichment (IDT Technologies), then sequenced on the Illumina HiSeq platform. These captured data were produced for baseline and surgical samples from 19 of the 22 WGS cases, and for an additional 38 cases. SNVs and Indels were called as described in the Supplementary Methods.

**RNA library preparation and sequencing.** RNA preparation was performed according to the Illumina TruSeq mRNA protocol with Poly-A selection. Unstranded paired-end sequencing was performed on the Illumina HiSeq 2000 platform, producing $48 \times 7 \times 48$ paired-end reads.

**Expression profiling.** RNA-seq data were aligned with Tophat v2.0.8 (de novo mode, reference only, reference guided) and expression levels calculated with Cufflinks v2.1.1 (params: --max-bundle-length = 10,000,000). RNAseq read-counts for all coding mutations were calculated using bam-readcount 0.7 (https://github.com/genome/bam-readcount). The samples were clustered by applying unsupervised hierarchical clustering on a matrix of sample-by-sample correlations derived from gene expression (FPKM) values (Supplementary Data 7 and 8).

**Clonality and clonal instability.** The clonal architecture of each tumour was inferred with sciClone (version 1.0.7; ref. 22), using default parameters except for copyNumberMargins set to 0.25. The median VAF of each subclone was used as input for clonevol (https://github.com/hdng/clonevol), which constructed evolutionary trees and estimated the absolute percentage of each clone in each sample. The clonal instability index was calculated as the mean of absolute differences between pre and post clonevol-inferred clonal percentages. A value of 0 represents no change, and 1 represents completely distinct tumours. In this study, 0.1 was used as an arbitrary cutoff between 'stable' and 'dynamic' tumours. BRC17 and BRC48 had complex patterns of copy number and ploidy changes preventing automated clonal inference, but clearly fall into this classification, with mutations specific to both baseline and surgical tumours. BRC18 did not have clear separation between subclonal populations, but contained a large number of mutations that were present in roughly half of the baseline sample tumour cells and were significantly enriched (and mostly expressed) in the surgical sample tumour cells.

Links to all the data and scripts used for clonal inference are found in Supplementary Software 1.

**Code availability.** Code used for analyses is all open source and detailed versions and parameters are provided above.

**Data availability.** The baseline whole-genome sequences have already been reported[20] and deposited in dbGaP with accession ID phs000472. Both the RNA sequencing data and the genomic sequencing data from the post-treatment samples and extension cohort have been added to the same study. All other data are contained within the Article or Supplementary Information files, or available from the authors upon request.

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

## Acknowledgements

Research reported in this publication was supported by the National Cancer Institute of the National Institutes of Health under Award Numbers U10CA180821 and U10CA180882 (to the Alliance for Clinical Trials in Oncology), and the following grants: 5U10CA180833 and 5U10CA180858. The content is solely the responsibility of the authors and does not necessarily represent the official views of the National Institutes of Health. Also supported in part by funding provided by grants U54HG003079 from the National Human Genome Research Institute to R.K.W., R01-CA095614 to M.J.E., U24-CA114736, U10-CA076001, and U01-CA114722 from the National Cancer Institute; by the Breast Cancer Research Foundation; Komen Promise Grant PG12220321 to M.J.E., a Komen St Louis Affiliate Clinical Trials Grant; and support for Z1031 from Pfizer and Novartis. M.J.E. is a McNair Medical Foundation Scholar and the recipient of a Cancer Prevention Research Institute of Texas established investigator award RR140033. We thank the participants of the clinical trials for their critically important samples provided for the study.

## Author contributions

C.A.M., R.K.W., M.J.E. and E.R.M. designed the experiments and data analysis. C.A.M., Y.G., C.L., O.L.G., M.G., D.S., T.L. and D.E.L. performed the data analysis. M.J.E. led the clinical investigations and biomarker analysis. C.A.M., Y.G. and C.L. prepared the figures and tables. V.J.S. provided statistical support. M.W. and K.D. provided pathology analysis. K.H., J.S., T.W., G.A.C. and S.R.D. provided study materials, patients or clinical support. C.A.M., E.R.M. and M.J.E. wrote the manuscript. Y.G., S.R.D., R.K.W. and D.E.L. critically read and commented on the manuscript.
