## [Peer review file · Nature Communications]

Reviewers' Comments:

Reviewer #1 (Remarks to the Author)

This manuscript reports genome sequencing and expression analysis data for 22 primary ER+ breast tumours at baseline and after 4 months of neo-adjuvant aromatase inhibitor treatment. The aim of the study was to annotate the innate molecular features that can predict response to AI treatment and to map the changes that occur throughout the treatment phase. Understanding the molecular event that can predict response to AI treatment has clinical application by identifying women who may benefit from alternative treatments (assuming there are biomarkers that can confidently predict this).

This manuscript is descriptive and given the relatively small sample size (albeit comprehensive analysis of each sample) the conclusions are also necessarily rather generalized. Some of the data presented has been previously published but the authors have performed additional sequencing on pre and post treatment samples. There are no specific mutations or profiles identified which can predict response to AI. The overall conclusion of the study is that the genomic landscape of these tumors is generally complex and they are often very heterogeneous and therefore caution needs to be applied when making clinical decisions based on a single biopsy. One interesting observation is that some patients with an initial diagnosis of an ER+ tumor who then develop an ER- tumor are actually due to a second, occult ER- tumor that was not diagnosed at presentation.

Relative to the conclusions, the manuscript is overly long with much of the data being repeated in the discussion. It would be better to summarize the genomic data in a table with only the highly relevant features highlighted in the text.

Specific comments.

1. Some metrics regarding the depth of coverage of the sequencing need to be provided in the text rather than referring to a previous publication (in any cases not all the data has been published before). Were tumors microdissected and if not what is the estimated tumor proportion.
2. It is unclear if any Sanger validation was performed on some of the key somatic mutations or were all assumed to be genuine based solely on the NGS pipeline?
3. The validation targeted gene panel contained 83 genes from the literature but were genes from the ER+ pre and post treatment cases included?

Reviewer #2 (Remarks to the Author)

This paper reports a comprehensive analysis of a small group of postmenopausal estrogen receptor (ER) positive breast cancers before and after 4 months of neoadjuvant aromatase inhibitor.

Its strengths are the ability to evaluate molecular characterizations of early stage breast cancer before and after a substantial period of therapy when information about clinical response is known (as opposed to a window trial where duration of therapy is very short or studies in the metastatic setting) and the intensive nature of the genomic analyses undertaken by a highly experienced group.

Several comments follow:

In the abstract, it would be prudent to modify the last sentence away from "... a naïve proposition" to a more modulated wording.

How exactly were the cases for study selected from the parent trial(s)? This is important as we try to generalize even from this small set of tumors to the larger population of patients. Are these tumors/patients representative of the larger studies or were they selected/enriched in some fashion?

Were all studies done on cores taken from the pre-treatment primary and the post-treatment surgical specimen? If yes, were there any guiding principles about how the cores were obtained-eg center of the tumor, periphery of the tumor? Given the results described, it appears that location may be important.

More information about how to define "clonal instability" would be useful. How is "complex" defined? Is there a specific cutoff on the 0 to 1 scale for clonal instability that denotes the cutoff between simple and complex? Is this an important concept if there is no correlation between clonal instability and AI response (at least as judged by change in Ki67 or Ki67 at surgery)? Is there any information about the initial T size of the tumors that were studied or any (even anecdotal) information about clinical response (realizing that major clinical responses including cCR are not going to be the norm with 4 months of endocrine therapy)?

This reviewer was surprised that so many clonally complex tumors switched intrinsic subtype from luminal B to luminal A under the pressure of AI therapy. One might have thought that the opposite would be the case-that luminal B would be less endocrine responsive and therefore likely to emerge during therapy.

We thank the reviewers for their close reading of the manuscript. Responses to their comments are below.

Reviewer 1:

Relative to the conclusions, the manuscript is overly long with much of the data being repeated in the discussion. It would be better to summarize the genomic data in a table with only the highly relevant features highlighted in the text.

We have shortened the manuscript's results section, and the discussion section has also been condensed to reduce redundancy with the results.

1. Some metrics regarding the depth of coverage of the sequencing need to be provided in the text rather than referring to a previous publication (in any cases not all the data has been published before). Were tumours microdissected and if not what is the estimated tumor proportion.

The tumors were not microdissected, so the tumor purity is indeed variable. A new Supplemental Table 4 has been added which contains coverage metrics and purity estimates, and this has been referenced in the first paragraph of the results section.

2. It is unclear if any Sanger validation was performed on some of the key somatic mutations or were all assume to be genuine based solely on the NGS pipeline?

The days of using Sanger validation to support the accuracy of our pipeline are long past, as we (and others) have demonstrated the superiority of NGS to Sanger methods. Our well-established approach to validating variants in this study revolves around initial detection with WGS then design of custom capture arrays that allow us to obtain deeper sequencing over the regions of interest. This is followed by a mutation calling pipeline that has been extensively tested and applied to tens of thousands of tumors. As a final check, each site is evaluated by an experienced manual reviewer to cull the few artifacts that are not removed algorithmically. While coverage and purity limit our sensitivity to low-VAF mutations in many cases, the specificity of these sites should be very high.

3. The validation targeted gene panel contained 83 genes from the literature but were genes from the ER+ pre and post treatment cases included?

There were several rounds of validation, and we have made this more clear in the Methods section. The 22 WGS samples from the original paper were validated using custom capture arrays targeted to the mutations in each tumor.

Later, after sequencing the post-AI samples, a second round of custom capture validation was performed on 77 samples from these 22 patients, with this array based on the mutations discovered in both the baseline and surgical samples. Finally, we assayed a larger cohort with the 83-gene targeted panel (19 patients that also had WGS, plus 38 additional individuals). This 83-gene panel was designed to cover the exons of the most commonly mutated genes in Breast Cancer, as derived from the TCGA study and others.

Reviewer #2 (Remarks to the Author):

In the abstract, it would be prudent to modify the last sentence away from "... a naïve proposition" to a more modulated wording.

We have replaced this sentence with:

"The observed clonal complexity of the ER+ breast cancer genome suggests that precision medicine approaches based on genomic analysis of a single specimen are likely insufficient to capture all clinically significant information."

How exactly were the cases for study selected from the parent trial(s)? This is important as we try to generalize even from this small set of tumors to the larger population of patients. Are these tumors/patients representative of the larger studies or were they selected/enriched in some fashion?

There was an unbiased selection. All samples were selected based upon having appropriate consents, with an estimated minimum 70% tumor content (by nuclei), and with DNA available from both the baseline and surgical time points. WGS was performed on the 22 cases that met these criteria and had a previously sequenced baseline sample (chosen in a similarly unbiased manner). The targeted extension cohort consisted of all additional samples that met the criteria for consent, tumor content and DNA availability listed above.. The selection criteria have been clarified in the first paragraph of the Methods section.

Were all studies done on cores taken from the pre-treatment primary and the post-treatment surgical specimen? If yes, were there any guiding principles about how the cores were obtained-eg center of the tumor, periphery of the tumor? Given the results described, it appears that location may be important.

All cores were taken according to the study protocol, but the locations would have been essentially random due to two factors: The first is that when dealing with fresh tissue biopsies, the center and periphery of the tumor are only grossly identifiable by visual inspection. The second is that this was a multi-center trial and there were no specific guidelines about biopsy location communicated to the many surgeons through the study protocol. Thus, this study cannot answer

questions about the spatial organization of the tumors (nor was it designed to). Information to this effect has been added to the supplement.

More information about how to define "clonal instability" would be useful. How is "complex" defined? Is there a specific cutoff on the 0 to 1 scale for clonal instability that denotes the cutoff between simple and complex? Is this an important concept if there is no correlation between clonal instability and AI response (at least as judged by change in Ki67 or Ki67 at surgery)?

"Simple" and "complex" designations were assigned based on the number of subclones (no subclones vs one or more subclones). We have also added to the text that 0.1 was used as the arbitrary cutoff between "stable" and "dynamic" in this study. While instability does not correlate directly with AI response, this is perhaps unsurprising, as tumors may either contain "innate" resistance in the founding clone, or may harbor a rare subclone that resists treatment and expands into the dominant clone at surgery. In both cases, the Ki67 would remain high, but the clonal instability would be low in the former and high in the latter. So while instability is not predictive of response, it does reveal something about the mechanism and origin of therapy resistance in a tumor.

Is there any information about the initial T size of the tumors that were studied or any (even anecdotal) information about clinical response (realizing that major clinical responses including cCR are not going to be the norm with 4 months of endocrine therapy)?

We agree that this would be interesting, but this information was not collected as part of the multi-center trial from which we obtained these samples.

This reviewer was surprised that so many clonally complex tumors switched intrinsic subtype from luminal B to luminal A under the pressure of AI therapy. One might have thought that the opposite would be the case—that luminal B would be less endocrine responsive and therefore likely to emerge during therapy.

This switching was somewhat expected, as the difference between Luminal A and Luminal B subtypes is driven by the proliferation score derived from gene expression (A = low, B = high). In every single tumor that switched from B to A, the Ki67 was reduced at surgery, and in most cases (8/10), the tumor was classified as AI-sensitive. In order for the opposite to occur (a change from A to B), there would have to be a switch to estrogen-independent proliferation. Events that would drive such a change are not common, and would be unexpected in this relatively short timeframe of four months.

Reviewers' Comments:

Reviewer #1 (Remarks to the Author)

This revised manuscript is much improved and more clearly articulates the data. I am satisfied with the alterations made with the exception of the response to the question about the Sanger Validation. It most certainly is not the practice of all the experienced NGS laboratories that I am associated with that Sanger Validation has been superseded by the bioinformatics pipeline. Even the best pipelines have errors and even manual inspection cannot unequivocally call all variants. The manuscript needs to at least describe the in silico validation process and explicitly state that Sanger validation was not performed.

Reviewer #2 (Remarks to the Author)

The authors have satisfactorily addressed the key elements of the review.

Reviewer #1 (Remarks to the Author):

This revised manuscript is much improved and more clearly articulates the data. I am satisfied with the alterations made with the exception of the response to the question about the Sanger Validation. It most certainly is not the practice of all the experienced NGS laboratories that I am associated with that Sanger Validation has been superseded by the bioinformatics pipeline. Even the best pipelines have errors and even manual inspection cannot unequivocally call all variants. The manuscript needs to at least describe the in silico validation process and explicitly state that Sanger validation was not performed.

While we understand that Reviewer #1 is concerned about possible false positives in the data we present here, we believe that their request that we perform Sanger validation is unreasonable. As we pointed out in our original response to their concerns, the days of using Sanger validation to support the accuracy of our pipeline are long past, as we (and others) have demonstrated the superiority of NGS to Sanger-based methods for detecting somatic variants in cancer genomes. We can point to literally hundreds of papers from dozens of research groups published in the last five years that do not perform Sanger validation of their calls. We also note that we report well over 40,000 variants in this study, making large-scale Sanger validation infeasible.

Our well-established approach to validating variants in this study revolves around initial detection with WGS then design of custom capture arrays that allow us to obtain deeper sequencing over the regions of interest. This is followed by a mutation calling pipeline that has been extensively tested and applied to tens of thousands of tumors.

As part of the CAP/CLIA certification process, we recently performed extensive validation of our core bioinformatics pipelines, which use the same variant callers and very similar parameters to the calling performed in this study. In tests using over 10,000 gold-standard mutations from HapMap samples, diluted to provide a range of allele frequencies, we had a positive predictive value of over 90% for calls greater than 5% VAF. When these pipelines are coupled with extensive manual review from individuals with nearly ten years of experience in next-gen sequencing quality control, we feel very confident that the vast majority of calls that we present in the paper are true-positives.

We have added the following text to the methods, which provides a bit more information about the expected performance of our variant-calling pipeline and review process:

Controlled tests of our somatic mutation calling pipeline (with similar configuration to that used here) have established a positive predictive value (PPV) of over 90% for mutations down to VAFs of ~5%, so we expect the

baseline false-positive rate to be low in these data. Results from the validation sequencing pipeline were then subjected to extensive manual review to validate the presence of variants and remove additional false-positive somatic calls. This involved manually inspecting sequencing reads using the Integrated Genomic Viewer (IGV), and flagging a number of different types of sequencing artifacts, including: insertions and deletions at the ends of homopolymer runs, calls that were present in the matched normal sample (or other normal samples), and regions with substantial numbers of low-or-zero quality mapping scores, indicating potential reference build problems or mapping artifacts. A similar process was used to inspect copy number calls, removing those resulting from alignment errors in highly repetitive regions of the genome or that span assembly gaps or centromeres